# Long-range exciton transport and slow annihilation in two-dimensional hybrid perovskites

Shibin Deng [1,4], Enzheng Shi [2,4], Long Yuan [1], Linrui Jin[1], Letian Dou [2,3] & Libai Huang [1]*

Two-dimensional hybrid organic-inorganic perovskites with strongly bound excitons and tunable structures are desirable for optoelectronic applications. Exciton transport and annihilation are two key processes in determining device efficiencies; however, a thorough understanding of these processes is hindered by that annihilation rates are often convoluted with exciton diffusion constants. Here we employ transient absorption microscopy to disentangle quantum-well-thickness-dependent exciton diffusion and annihilation in two-dimensional perovskites, unraveling the key role of electron-hole interactions and dielectric screening. The exciton diffusion constant is found to increase with quantum-well thickness, ranging from $0.06 \pm 0.03$ to $0.34 \pm 0.03$ cm$^2$ s$^{-1}$, which leads to long-range exciton diffusion over hundreds of nanometers. The exciton annihilation rates are more than one order of magnitude lower than those found in the monolayers of transition metal dichalcogenides. The combination of long-range exciton transport and slow annihilation highlights the unique attributes of two-dimensional perovskites as an exciting class of optoelectronic materials.

[1] Department of Chemistry, Purdue University, West Lafayette, IN 47907, United States. [2] Davidson School of Chemical Engineering, Purdue University, West Lafayette, IN 47907, United States. [3] Birck Nanotechnology Center, Purdue University, West Lafayette, IN 47907, United States. [4] These authors contributed equally: Shibin Deng, Enzheng Shi. *email: libai-huang@purdue.edu

Two-dimensional (2D) Ruddlesden-Popper phase organic–inorganic metal halide perovskites have emerged as an exciting class of optoelectronic materials. Efficient light emitting diodes (LED), lasers, and solar cells with superior stability over their 3D counterparts have been demonstrated[1–11]. 2D perovskites have a general structure of $(RNH_3)_2(A)_{n-1}B_nX_{3n+1}$, where R is an alkyl or aromatic moiety, A is an organic cation, B is a metal cation, and X is a halide[12–14]. The variable $n$ corresponds to the number of the inorganic quantum-well layers sandwiched between the long organic chains $RNH_3$. Because dielectric screening is low from the surrounding organic ligands, excitons with binding energy as high as 0.5 eV dominate the optical properties of these materials[15–20]. The exciton binding energy for the $n = 1$ structure is comparable with those found in monolayer transition metal dichalcogenides (TMDCs)[18,21]. The structure of 2D perovskites is extremely programmable; for instance, continuous tuning of the exciton emission over a wide range of energy can be achieved by either changing $n$ or the halide in the composition[11,22]. In addition, both the R and A site cations can be modified to influence electron-phonon coupling, dielectric environment, and lattice disorder[3,19,23]. The highly tunable structure coupled with emission quantum yield as high as 79%[3] represents a significant advantage of 2D perovskites over other 2D semiconductors.

To harvest excitons in a device, a thorough understanding of exciton transport and annihilation in 2D perovskites is necessary. For example, the ability to transport excitons over long distances is necessary for solar cell performance. On the other hand, many-body exciton–exciton annihilation (EEA), a form of Auger recombination, is an important loss mechanism that limits the density of excitons and ultimately determines the efficiency of lasers and LEDs. EEA occurs when an exciton recombines non-radiatively by transferring its kinetic energy to another exciton. Rapid EEA rates are usually associated with nanostructures with large exciton binding energies because of the enhanced electron-hole interactions in these systems[24–26], representing a key challenge in realizing their optoelectronic applications.

Despite the impressive progress made in the optoelectronic devices based on 2D perovskites[11], the understanding of exciton transport is extremely limited and currently there is no direct measurement on exciton diffusion. Because of the requirement of two excitons in proximity, exciton diffusion typically proceeds annihilation in 1D or 2D materials. EEA can occur either in the diffusion-limited or the reaction-limited regime[27]. To differentiate these two regimes, measurements of exciton population dynamics in both spatial and temporal domains are required. However, the majority of studies on exciton annihilation so far are based on time-resolved photoluminescence (PL) or transient absorption (TA) spectroscopy that offers no spatial resolution, leading to annihilation rates convoluted with exciton diffusion constants[28–30], making it difficult to elucidate factors that control these two processes independently.

To address this challenge, here we employ transient absorption microscopy (TAM) as a direct means to image exciton population in space and in time to disentangle exciton diffusion and annihilation in 2D perovskites. We have demonstrated in a previous work the capability of TAM in imaging free carrier diffusion and extracting Auger recombination constants in bulk 3D perovskites[31]. Distinct from the results in 3D perovskites, the measurements on 2D perovskites elucidated the critical role of electron-hole interactions in controlling exciton dynamics and transport. These results showcase the unique ability of 2D perovskites in achieving a combination of large exciton binding energy, long-range exciton transport, and slow annihilation, suggesting their large potential in optoelectronic applications.

## Results

**Structural and optical characterizations**. 2D perovskites with composition of $(C_4H_9NH_3)_2(CH_3NH_3)_{n-1}Pb_nI_{3n+1}$ ($(BA)_2(MA)_{n-1}Pb_nI_{3n+1}$) were investigated in this study, as schematically illustrated in Fig. 1a, b. Pure-phase single crystals with $n$ ranging from 1 to 5 were mechanically exfoliated on fused silica for the optical measurements as shown in Supplementary Fig. 1. The single-crystalline samples allow for the elucidation of the role of dielectric screening by systematically investigating the dependence on quantum-well thickness $n$. In these samples, the $[PbI_6]^{4-}$ octahedral layers are parallel to the substrate surface, as shown in the X-ray diffraction profiles (Supplementary Fig. 2). In order to correctly determine the exciton densities for the optical measurements, we carried out microreflectance and transmittance spectroscopy as well as atomic force microscopy (AFM) measurements (Supplementary Fig. 3). By solving the Fresnel's equations, we obtained the linear extinction coefficient $\alpha$ (Fig. 1c) and the refractive index (Supplementary Fig. 4) as detailed in Supplementary Note 1.

Excitons dominate the optical responses in all five structures as evidenced in the absorption and PL spectra. In the absorption spectra, narrow exciton resonances to the lower-energy side of the broad continuum absorption band are observed (Fig. 1c). Sharp exciton emission peaks correlated with the exciton absorption resonances are seen in the PL spectra (Fig. 1d), indicating high quality and phase-purity of the single crystals. These 2D perovskite quantum wells are very efficient absorbers with extinction coefficient $\alpha$ on the order of $10^5$ cm$^{-1}$ (Fig. 1c), resulting from the strongly bound excitons with large oscillator strengths, similar to TMDCs[32]. Exciton binding energy $E_b$ for $n = 1$ is found to be as large as 470 meV[18], with a Bohr radius $a_0$ of around 1 nm[15,16]. As shown in Fig. 1c, exciton oscillator strength decreases as $n$ increases, which can be explained by that the oscillator strength is proportional to $E_b$[18,33]. $E_b$ has been reported to reduce to 125 meV when $n$ increases to 5[18]. The integral $\alpha$ per Pb atom converge at high energy side of the spectra (Supplementary Fig. 5). This is because the sum of the oscillation strength over the entire spectral range is determined by the number of electrons involved in the transition, which should be the same for each Pb atom in $[PbI_6]^{4-}$. In other words, due to the stronger bound excitons, more oscillator strength is concentrated in the exciton absorption band for the $n = 1$ than for the $n > 1$ quantum wells. The exciton resonance, as given by $E_{sp} - E_b$ ($E_{sp}$ is the single-particle bandgap), shifts to higher energy as $n$ decreases (Figs. 1c, d). This is due to the quantum confinement along the out-of-plane direction which gives rise to the enlarging $E_{sp}$ with a decreasing $n$ value. The enlarged $E_{sp}$ is counteracted to some extent by the increased $E_b$.

**TAM imaging of exciton transport**. We employed TAM to image the exciton population dynamics in both spatial and temporal domains (more details in Supplementary Fig. 6 and Methods). We have recently demonstrated TAM as an effective tool for imaging exciton transport in 2D semiconductors[34,35]. In these measurements, the exciton population was monitored by probing resonantly at the photoinduced bleach of the exciton bands. Figure 2a shows the representative TA spectra for $n = 5$, and the photoinduced bleach band at 1.86 eV is consistent with the exciton absorption resonance in Fig. 1c. The probe beam was scanned relative to the pump in space using a Galvanometer scanner to image exciton transport. 2D TAM images were obtained by plotting pump induced change in probe transmission $\Delta T$ as a function of probe position. At time zero, initial exciton distribution was given by a Gaussian pump beam with a pulse duration of approximately 300 fs, $N(x, y, 0) = N_0 \exp\left[-\frac{(x-x_0)^2}{2\sigma_{x,0}^2} - \frac{(y-y_0)^2}{2\sigma_{y,0}^2}\right]$. TAM images at

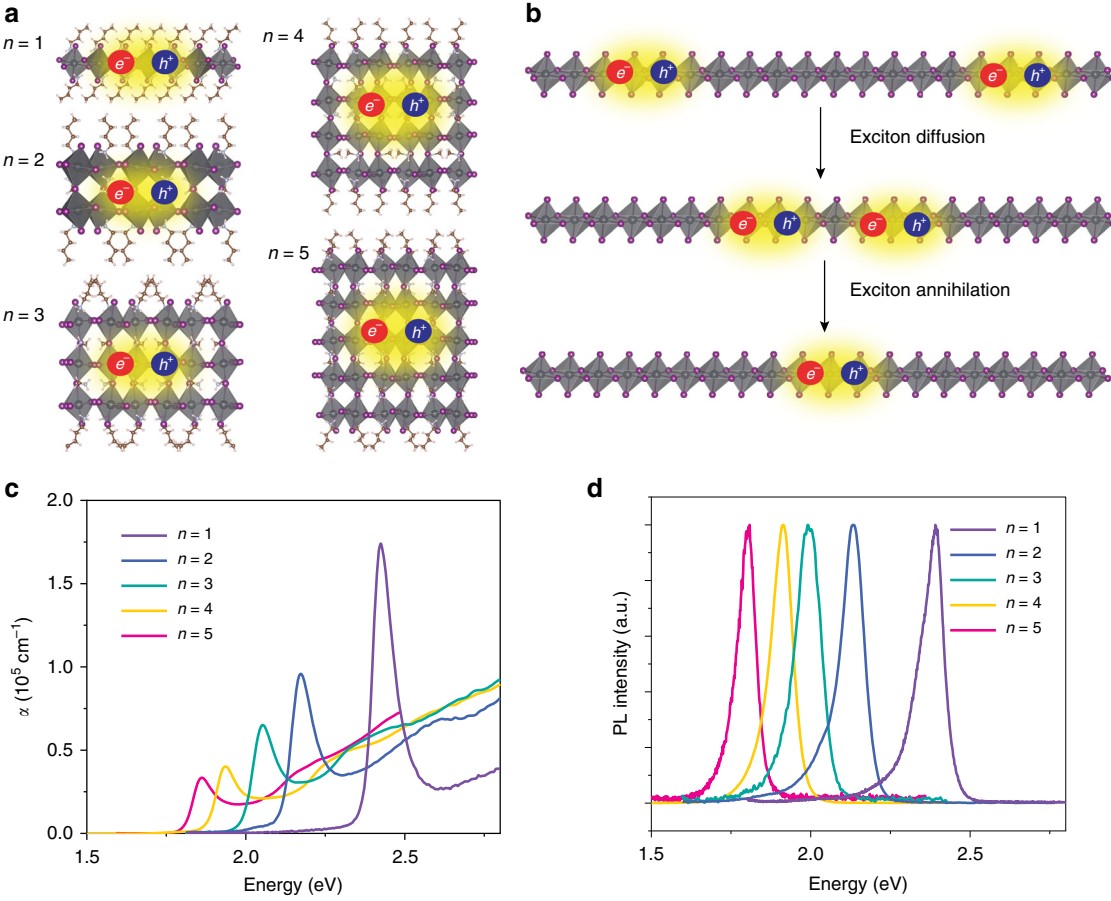

**Fig. 1 The structure and optical characterizations of 2D perovskite quantum wells. a** Illustration of the crystal structures and excitons of $(BA)_2(MA)_{n-1}$ $Pb_nI_{3n+1}$ with $n$ varying from 1 to 5. **b** Scheme of exciton diffusion and annihilation. **c** Linear extinction coefficient $\alpha$ and **d** PL spectra of $(BA)_2(MA)_{n-1}Pb_nI_{3n+1}$ with $n$ varying from 1 to 5.

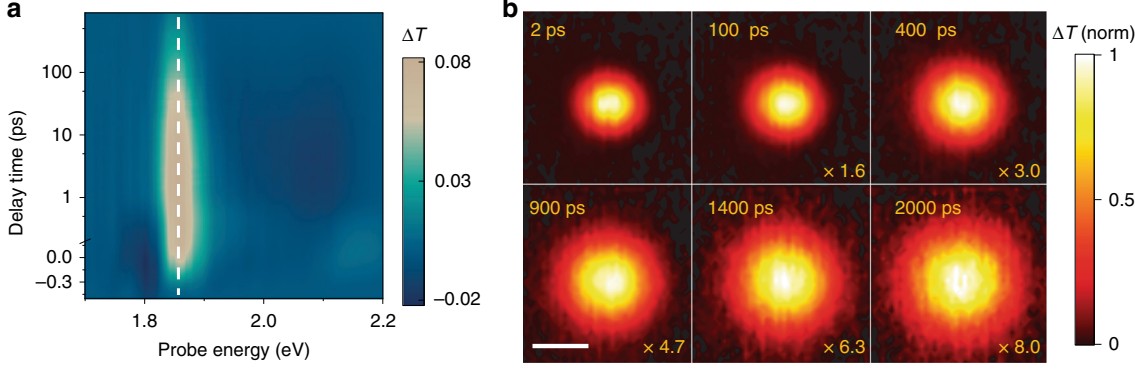

**Fig. 2 Transient absorption spectroscopy and microscopy for $n = 5$. a** TA spectra for $(BA)_2(MA)_4Pb_5I_{16}$, showing the photoinduced bleaching of the exciton resonance at 1.86 eV. **b** Two-dimensional TAM imaging of $(BA)_2(MA)_4Pb_5I_{16}$ with pump-probe delay time varying from 2 ps to 2 ns. Pump beam is fixed while probe beam scanning to form the images. Pump photon energy: 1.97 eV, probe photon energy: 1.86 eV. Excitation density: $2.4 \times 10^{12}$ cm$^{-2}$. Maximum intensities are normalized to 1 by multiplying with the factors shown in the images. Scale bar: 1 µm.

later delay times measured the exciton diffusion away from the initial location that results in a population distribution of

$$N(x, y, t) = N_t \exp\left[ -\frac{(x-x_0)^2}{2\sigma_{x,t}^2} - \frac{(y-y_0)^2}{2\sigma_{y,t}^2} \right].$$

The exciton population was observed to expand as pump-probe delay time increased. The representative TAM images for $n = 5$ at time delays ranging from 2 ps to 2 ns are illustrated in Fig. 2b. These TAM measurements reflect the in-plane exciton transport in

the quantum wells because the $[PbI_6]^{4-}$ octahedral layers are parallel to the substrate surface. The exciton diffusion in ab plane is isotropic so that $\sigma_{x,t} = \sigma_{y,t} = \sigma_t$. To reduce the data redundancy, one-dimensional scanning $N(x, 0, t)$ instead of two-dimensional imaging $N(x, y, t)$ was performed for the following TAM measurements. 1D imaging also allowed more measurement cycles to improve the signal-to-noise ratio. Although the pump and probe beam sizes were diffraction-limited (100 s of nm in visible light

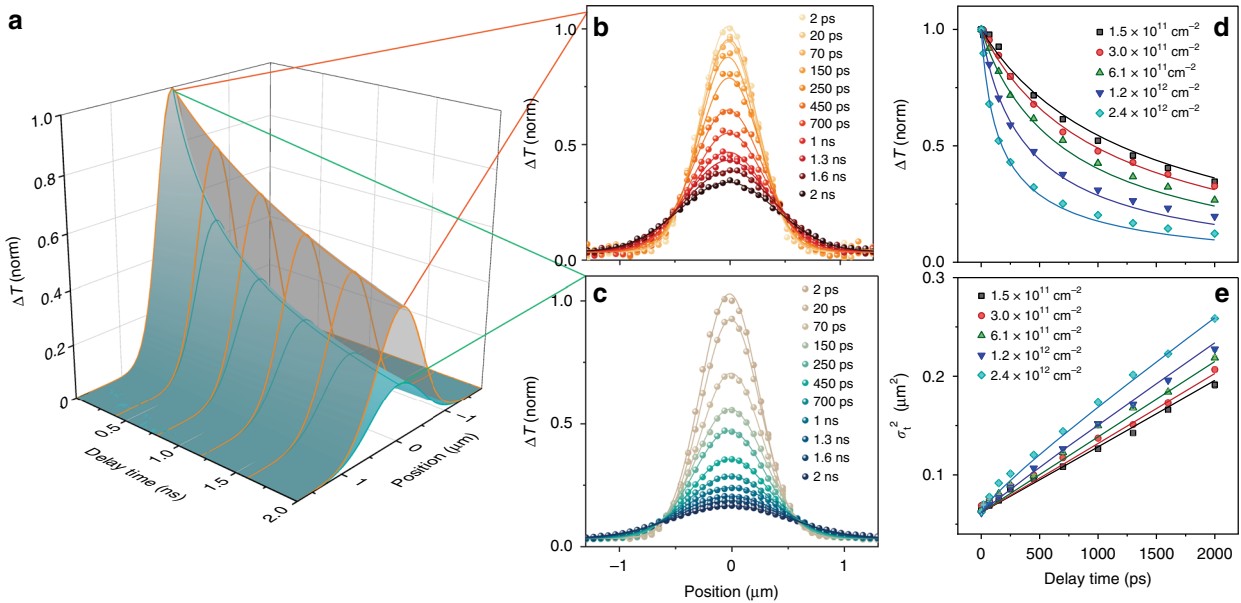

**Fig. 3 Exciton population dynamics for $n = 5$. a** 3D population surface $N(x, 0, t)$ simulated at initial exciton densities of $1.5 \times 10^{11}$ cm$^{-2}$ (low-density limit, gray surface) and $2.4 \times 10^{12}$ cm$^{-2}$ (high-density limit, blue surface). **b–c** Experimental exciton population spatial profiles at different delay time measured by TAM with initial excitation densities of $1.5 \times 10^{11}$ cm$^{-2}$ (**b**) and $2.4 \times 10^{12}$ cm$^{-2}$ (**c**). The solid lines are fits to a Gaussian function to obtain $\sigma_t^2$. **d–e** Experimental (symbols) and simulated (solid lines) $\Delta T$ at 0 pump-probe separation (**d**) and time evolution of $\sigma_t^2$ (**e**) with excitation densities ranging from $1.5 \times 10^{11}$ cm$^{-2}$ to $2.4 \times 10^{12}$ cm$^{-2}$. Pump photon energy: 1.97 eV, probe photon energy: 1.86 eV.

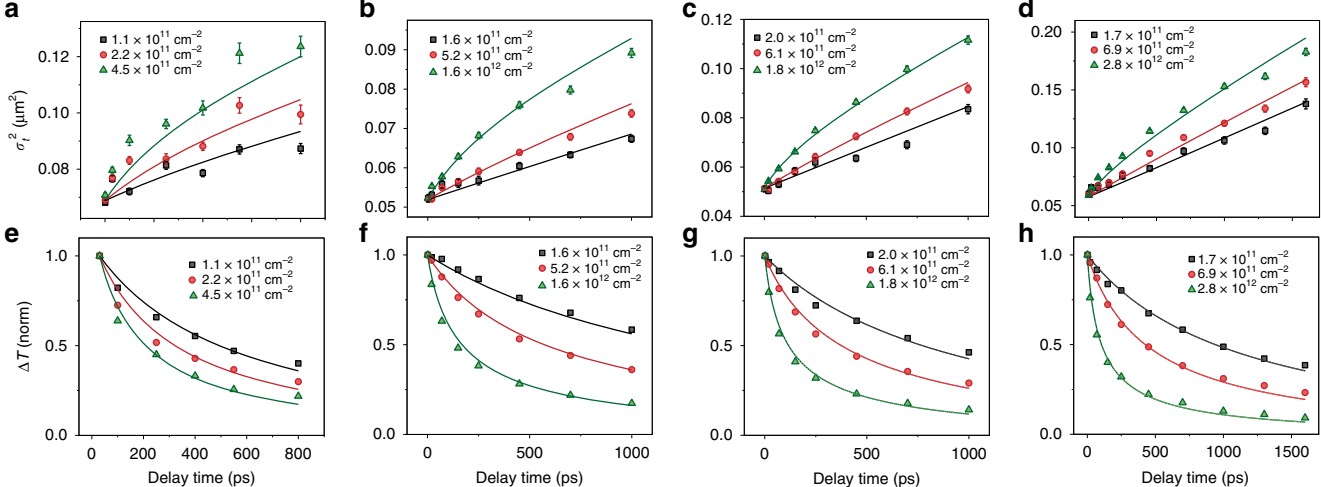

**Fig. 4 Exciton population dynamics for $n$ ranging from 1 to 4. a–d** Experimental (symbols) and fitted (solid lines) time evolution of $\sigma_t^2$. **e–h** Experimental and fitted time evolution of $\Delta T$ at 0 pump-probe separation. $n = 1$ (**a, e**), $n = 2$ (**b, f**), $n = 3$ (**c, g**), and $n = 4$ (**d, h**). Pump photon energies are 2.61 eV, 2.32 eV, 2.14 eV, and 2.14 eV for $n$ ranging from 1 to 4, respectively. Probe photon energies are 2.43 eV, 2.18 eV, 2.03 eV, and 1.97 eV for $n$ ranging from 1 to 4, respectively. The error bars indicate the standard error of Gaussian fit of the exciton population spatial profiles.

range), the accuracy to determine the width of exciton population profile was determined by the signal-to-noise ratio and it was better than 50 nm[31] (more details in Supplementary Note 2).

**Modeling of exciton diffusion and annihilation.** To extract both exciton diffusion constants and annihilation parameters, we carried out exciton-density-dependent TAM measurements. The initial exciton densities were kept below $2.4 \times 10^{12}$ cm$^{-2}$, well below the Mott density estimated by $a_0^{-2} = 10^{14}$ cm$^{-2}$; thus, excitons remained the main excited-state species. All the measurements were performed in the linear regime so that the TA signal was proportional to exciton density (Supplementary Fig. 7), i.e., $\Delta T \propto N$. When exciton–exciton annihilation is negligible at low exciton densities, the time and spatial dependent exciton density is given by,

$$\frac{\partial N(x, y, t)}{\partial t} = D\left[\frac{\partial^2 N(x, y, t)}{\partial x^2} + \frac{\partial^2 N(x, y, t)}{\partial y^2}\right] - k_1 N(x, y, t) \quad (1)$$

where $D$ is the diffusion constant, $k_1$ is the exciton recombination rate. Under this condition, the experimentally measured $\sigma_t^2 - \sigma_0^2$ corresponds to the mean squared distance travelled by the excitons, and the diffusion constant $D = \frac{\sigma_t^2 - \sigma_0^2}{2t}$.

At high densities, the spatial-temporal population distribution reflects both exciton diffusion and annihilation. EEA is a bimolecular process, which is represented by an additional second-order term in the rate equation. Further, third-order

**Table 1 Exciton diffusion constant and recombination rate constants for $(BA)_2(MA)_{n-1}Pb_nI_{3n+1}$.**

| Structure | $D$ (cm$^2$s$^{-1}$) | $k_1$ (ns$^{-1}$) | $k_2$ ($10^{-3}$ cm$^2$s$^{-1}$) | $k_3$ ($10^{-14}$ cm$^4$s$^{-1}$) | $k_2/D$ |
|---|---|---|---|---|---|
| $n = 1$ | 0.06 ± 0.03 | 0.45 ± 0.08 | 17 ± 5 | – | 0.30 |
| $n = 2$ | 0.07 ± 0.03 | 0.23 ± 0.06 | 4.1 ± 1.0 | 0.18 ± 0.05 | 0.058 |
| $n = 3$ | 0.15 ± 0.04 | 0.30 ± 0.07 | 3.5 ± 0.6 | 0.25 ± 0.06 | 0.023 |
| $n = 4$ | 0.25 ± 0.06 | 0.2 ± 0.04 | 2.8 ± 0.6 | 0.11 ± 0.03 | 0.011 |
| $n = 5$ | 0.34 ± 0.03 | 0.15 ± 0.03 | 1.3 ± 0.3 | 0.11 ± 0.03 | 0.0038 |

The error bars are estimated from the sensitivity of model to the parameters. The measurement of 5 flakes for each $n$ value are repeatable within the error range.

Auger recombination of free charge carriers is also possible for $n > 1$ quantum wells, as a finite free carrier population exists due to smaller exciton binding energy[20]. Therefore, there are two additional terms to Eq. 1,

$$\frac{\partial N(x,y,t)}{\partial t} = D\left[\frac{\partial^2 N(x,y,t)}{\partial x^2} + \frac{\partial^2 N(x,y,t)}{\partial y^2}\right] \quad (2)$$
$$- k_1 N(x,y,t) - k_2 N^2(x,y,t) - k_3 N^3(x,y,t)$$

where $k_2$ describes EEA rate and $k_3$ corresponds to the Auger recombination rate of free carriers.

To understand the spatial-temporal behavior, we modeled the exciton population as function of pump-probe delay time and probe position using Eq. 2. As an illustration, Fig. 3a shows the 3D surfaces representing the temporal and spatial distribution of excitons at both low- and high-density limits, which is simulated by numerically solving Eq. 2 to obtain $N(x,0,t) = N_t \exp[\frac{-(x-x_0)^2}{2\sigma_t^2}]$, taking into account the convolution with the probe beam (~500 nm, full width at half maximum). Exciton population decays faster at higher densities as expected from more significant annihilation. Broadening of $\sigma_t^2$ is observed at high densities, which can be explained by faster EEA at the center compared to the edge of the pump beam due to the higher exciton density. Instead of plotting the spatial-temporal distribution of excitons $N(x,0,t)$ in 3D, we depicted $\sigma_t^2$ from the fitting of the Gaussian profiles (Fig. 3b, c) as well as the $\Delta T(0,0,t)$ at the center (0 pump-probe spatial separation) as a function of pump-probe delay time, which conveyed the same information. The fitting results are shown in Fig. 3b–e for $n = 5$. The experimental and simulated results for $n$ ranging from 1 to 4 quantum wells are illustrated in Fig. 4. The model has satisfactorily fitted the experimental results with the initial exciton density varying over one order of magnitude using the same set of parameters for each $n$. The initial exciton densities were determined experimentally using absorption coefficient shown in Fig. 1c. A deviation from the modeling by around 5% can be seen in the first 100 ps for $n = 1$, which is likely due to defect trapping of the excitons that is not included in the modeling. Defect density was the highest in $n = 1$, and probably not all the defects were passivated by the pump photons under our experimental conditions[30]. For $n$ ranging from 2 to 5, such deviation was not observed because defect density decreased in thicker quantum wells and the pump fluences used were sufficiently high to passivate the defects[30]. The extracted $D$, $k_1$, $k_2$, and $k_3$ from the fitting are summarized in Table 1. Sensitivity analysis of the model to the parameters is shown in Supplementary Fig. 8.

**The $n$ dependence of exciton diffusion and annihilation.** Both exciton diffusion constant $D$ and diffusion length $L_0$ exhibit strong dependence on $n$. As shown in Table 1 and Fig. 5a, $D$ increases as $n$ increases, from 0.06 ± 0.03 ($n = 1$) to 0.34 ± 0.03 cm$^2$ s$^{-1}$ ($n = 5$). Within the band transport picture[36], the diffusion constant is determined by $D = \frac{\tau_s kT}{m}$, where $\tau_s$ is the momentum

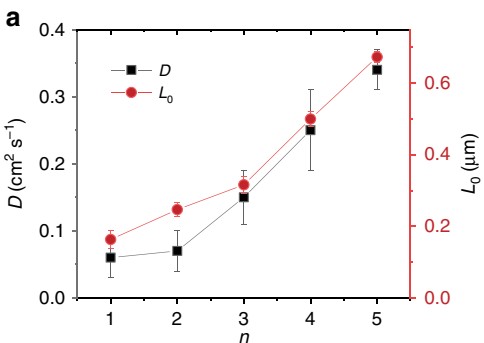

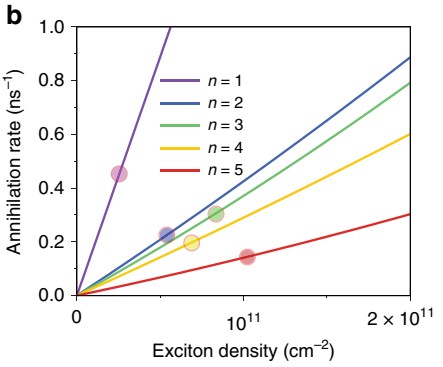

**Fig. 5 Exciton diffusion constant, diffusion length, and annihilation rate as a function of $n$. a** Exciton diffusion constant $D$ and diffusion length $L_0$ for $n$ ranging from 1 to 5. $D$ and $L_0$ increase as $n$ increases. The error bars are estimated from the sensitivity of model to the parameters. **b** Total annihilation rate as a function of $n$ for $(BA)_2(MA)_{n-1}Pb_nI_{3n+1}$, showing enhanced annihilation at decreasing $n$. As a comparison, the exciton recombination rates $k_1$ are denoted by the circles.

relaxation time corresponding to the inverse of the sum of all scattering rates, k is the Boltzmann constant, $T$ is temperature, and $m$ is the reduced mass of the exciton. $m$ was reported to be only weakly dependent on $n$[18], and therefore the $n$ dependence of $D$ is more likely resulting from the difference in $\tau_s$. The scattering sources include carriers, phonons, defects, and impurities. Defect density has been found to be higher for $n = 1$ than the thicker quantum wells[30], which contributes to a more frequent defect scattering (a shorter $\tau_s$) and a lower $D$. Dielectric screening could also play a role. As the $n$ value increases from 1 to 5, the thickness of inorganic layer grows from 0.6 to 3 nm leading to the effective optical-frequency dielectric constant enlarging from 4 to 5.5[37]. Because the scattering by charge carriers, polar optical phonons, and charge impurities are coulombic in nature, the reduced Coulomb interactions in the thicker quantum wells with higher dielectric constant results in a longer $\tau_s$. The exciton diffusion constants measured for the 2D perovskites are consistent with the carrier diffusion constant of around 1 cm$^2$ s$^{-1}$ at the 3D limit of $n = \infty$[38].

Long-range exciton transport over hundreds of nanometers is achieved in all five quantum-well structures. We calculate exciton diffusion length by $L_0 = \sqrt{2D/k_1}$ and $L_0$ as a function of $n$ is presented in Fig. 5a. $L_0$ increases from 160 to 670 nm as $n$ increases from 1 to 5 (Fig. 5a), due to the combined effects of higher diffusion constant and slower recombination rate $k_1$. The diffusion length in all five structures is on the same order or longer than the thickness of effective layer of solar cell devices[10].

It is possible that photon recycling that has been proposed to enhance carrier diffusion in 3D perovskites contributes to the long-range exciton transport observed here[39–41]. To differentiate the contributions from photon recycling from pure exciton diffusion, we have carried out Monte Carlo simulations as shown in Supplementary Fig. 9. As detailed in Supplementary Note 3, both electronic transport and photonic transport from photon recycling can be microscopically described as random walks but with significantly different parameters (Supplementary Table 1). For electronic transport, each step of the random walk during $\tau_s$ between scattering events is on the order of nm. To achieve a final average transport distance of 100 s of nm, excitons need to walk around $10^5$ steps randomly, resulting in a Gaussian distribution of transport distance. For photonic transport, the step size is much larger, on the order of 100 s of nm as determined by the penetration depth $\alpha^{-1}$. To achieve a final average transport distance of 100 s of nm, the number of steps for photons is small, such as 1 or 2, which results in a distribution of transport distance strongly deviates from the Gaussian type. Therefore, we can use the TAM profiles to determine if transport via photon recycling playing a role by examining the spatial distributions of the excitons. As shown in Supplementary Fig. 9, the exciton population profiles are described very well by Gaussian functions. Therefore, we conclude that photonic contributions from photon recycling in exciton transport is negligible.

EEA takes place at densities of around $10^{11}$ cm$^{-2}$ or lower, corresponding to an average exciton–exciton distance $d > 30$ nm, much larger than the exciton size of around 1–2 nm[15]. Thus, exciton diffusion must proceed annihilation. Note that Auger recombination of free carriers contributes significantly less than EEA at density of around $10^{11}$ cm$^{-2}$; thus, we focus our discussion on EEA rate $k_2$. We find that $k_2$ decreases with $n$; for $n = 5$, $k_2$ is determined to be $1.3 \times 10^{-3}$ cm$^2$ s$^{-1}$, which is an order of magnitude lower than that of $1.7 \times 10^{-2}$ cm$^2$ s$^{-1}$ for $n = 1$. For all five structures, exciton diffusion constants are significantly larger than annihilation rates (D $\gg k_2$). Therefore, EEA is reaction-limited instead of diffusion-limited, i.e., only a fraction of exciton encounters result in annihilation. We use the ratio of $k_2/D$ to describe the annihilation probability per encounter as given in Table 1. Although excitons are the most mobile in $n = 5$ and they encounter at the highest rate, only about 0.4% of the encounters result in annihilation. Annihilation probability increases by two orders of magnitudes to 30% for $n = 1$. The reaction-limited EEA in 2D perovskites is in contrast to the diffusion-limited EEA in other low-dimensional nanostructures, such as single-walled carbon nanotubes[27,42], which suggests that exciton diffusion and annihilation can be separately optimized.

The strong $n$ dependence of $k_2$ underscores the importance of electron-hole interactions and dielectric screening in exciton annihilation[26]. EEA is mediated by the coupling between the transition dipole for promoting exciton 1 to the higher excited state and that for exciton 2 to decay to the ground state[26]. As $n$ increases, the exciton wavefunction becomes located only in the inorganic layer with higher dielectric constant (Fig. 1a), which enhances dielectric screening. Therefore, for thicker quantum wells, the dipole-dipole interaction is weakened by more effective dielectric screening, leading to slower $k_2$. $k_2$ shares a similar $n$ dependence as the exciton binding energy $E_b$ because the dielectric screening also reduces $E_b$[24]. Another factor could also play a role is the spatial overlap between the initial and final exciton wavefunctions. As $E_b$ increase, exciton states contain components corresponding to electron and hole wavefunctions with large momenta, relaxing the requirement of momentum conservation[24]. For Auger recombination of free carriers, a maximum is observed for $k_3$ when $n = 3$. The nonmonotonic $n$ dependence of $k_3$ can be explained by the combination of the following two competing effects: (1) similar to EEA, the Auger recombination probability for free carriers per encounter decreases with increasing $n$ due to enhanced screening; and (2) the fraction of free carriers increases with as $n$ increases.

The observation of enhanced EEA rates at larger exciton binding energies in 2D perovskites are consistent with those reported in single-walled carbon nanotubes and colloidal quantum dots[24,26,42,43]. We note that there are a few previous studies on EEA and Auger recombination in 2D perovskites; however, an explicit $n$ dependence of EEA rate has not been reported. Delport et al.[30] reported EEA rate was nearly independent on $n$ using time-resolved PL spectroscopy. The discrepancy between our results and those reported in Delport et al.[30] could be due to that PL measurements only monitor a subset of the exciton population, namely the emissive species. In contrast, the photoinduced bleach TA signal reflects to all exciton population. Using TA spectroscopy, Chen et al.[29] reported both $k_2$ and $k_3$ increases as $n$ decreases; however, they interpreted the $k_2$ in terms of bimolecular recombination of free carriers instead of EEA. Auger recombination in mixed-phase 2D-3D perovskites has been studied by time-resolved THz measurements by Milot et al.;[28] however, THz measurements are sensitive to free carriers instead of excitons.

**Comparison with other 2D semiconductors**. We compare exciton transport and annihilation properties of 2D perovskites to those of TMDCs. We have measured the exciton diffusion constant in exfoliated WS$_2$ monolayers[34] to be around 2 cm$^2$ s$^{-1}$, more than one order higher than the $n = 1$ quantum-well. A possible reason is the formation of exciton polarons[44] that have larger effective mass leading to slower diffusion. Another feasible explanation is that structural disorder is much larger in hybrid perovskites than in TMDCs due to the ionic nature. Structural disorder results in fluctuations in the energy landscape, slowing down exciton transport[45]. We suggest that exciton transport in 2D perovskites can be further enhanced through structural tuning to increase rigidity and reduce structural disorder, for instance, by employing phenyl group motifs at the R site[3].

Notably, exciton annihilation is significantly reduced in 2D perovskites in comparison to TMDCs. In a previous work, we determined the exciton–exciton annihilation rate to be 0.41 cm$^2$ s$^{-1}$ for monolayer WS$_2$[46], similar to other TMDCs[47], which is more than one order of magnitude higher than that of $n = 1$. This large difference in exciton annihilation rates given similar exciton binding energies in two materials is intriguing and could be related to the unique dielectric function of hybrid perovskites, for instance, as the result of the motion of the polar organic cations[48]. Further, an advantage brought by the highly tunable structure of 2D perovskites is the ability to modulate Coulomb interactions to suppress annihilation through composition tuning. Different from TMDCs that undergo direct to indirect bandgap transition at bilayer[49], 2D perovskites are direct bandgap semiconductors with high PL quantum yields regardless of the value of $n$.

As a guide, we plot the total annihilation rates ($k_2N + k_3N^2$) as a function of exciton densities for $n$ ranging from 1 to 5 in Fig. 5b,

showing the reduction of annihilation loss by increasing $n$. $k_2$ becomes larger than $k_1$ at exciton density as low as $2 \times 10^{10}$ cm$^{-2}$ for $n = 1$, indicating that EEA turns into the dominant recombination pathway. This is probably one of the key reasons why high-performance solar cell and LED based on $n = 1$ 2D perovskites have not been successfully demonstrated so far. For $n = 5$, the switch over occurs at higher exciton density around $10^{11}$ cm$^{-2}$. The benefit of the reduction in annihilation is most significant for increasing from $n = 1$ to $n$ ranging from 2 to 4, suggesting that these structures with $E_b$ ranging from 150 to 250 meV might be optimal for light emitting devices with the combined benefits of large exciton binding energy and slow annihilation. This is consistent with a recent study[50] in which van-der-Waals-coupled 2D perovskite multi-quantum-wells were shown to have better light emitting performance than 3D perovskites due to strongly bound excitons. For solar cells, quantum wells with large $n$ are suitable because of the combined benefits of the small exciton binding energy and long-range diffusion. Further suppression of EEA should be possible through modulating R and A site organic cations to modify dielectric screening. Such promise has been demonstrated recently, for instance, a study has showed that Auger recombination rate in the hybrid perovskite nanocrystals containing polar organic cations is significantly lower than in the fully inorganic Cs-based ones[51].

## Discussion

A major challenge in realizing optoelectronic applications for low-dimensional nanostructures is to achieve long-range transport and suppressed annihilation while maintaining large exciton binding energy. As unraveled by the spatially- and temporally resolved measurements reported here, 2D perovskites present unique opportunities in addressing this challenge. The exciton binding energies of 2D perovskites are comparable to those found in TMDCs but their annihilation rates are orders of magnitude lower. These properties make 2D perovskites excellent candidates for light emitting applications. Further, 2D perovskites are also promising as solar cell materials, with long-range exciton diffusion over hundreds of nanometers. Finally, their structures are uniquely programmable, which allows for further enhancement of exciton transport and suppression of annihilation through composition engineering.

## Methods

**Synthesis of 2D halide perovskite crystals**. The 2D Ruddlesden-Popper phase halide perovskite crystals $(BA)_2(MA)_{n-1}Pb_nI_{3n+1}$ ($n$ ranging from 1 to 5) were synthesized via a solution precipitation reaction. PbO powders, hydroiodic acid (HI, 57 wt.% in $H_2O$), hypophosphorous acid ($H_3PO_2$, 50 wt. % in $H_2O$) were purchased from Sigma–Aldrich. Methylammonium iodide (MAI), formamidinium iodide (FAI), $n$-Butylammonium bromide (BABr), and $n$-Butylammonium iodide (BAI) were purchased from Greatcell Solar Ltd. All chemicals were used as received. Commercial Scotch tapes (3 M) were used to exfoliate 2D halide perovskite crystals. The mechanical exfoliation was performed in $N_2$-filled glovebox. The thicknesses of the flakes were measured by AFM (Veeco Dimension 3100). More details on the synthesis and sample preparation can be found in Supplementary Note 4.

**Optical absorption and photoluminescence (PL) measurements**. A home-built microscope was used for the micro-absorption and PL spectroscopy measurements. For the micro-absorption measurements, a halogen tungsten lamp was used as white light source, which was focused by objective on sample to form light spot with size of around 2 μm. The reflective or transmitted light was collected by an objective (×40, NA = 0.6, Nikon), and detected by a spectrometer (Andor Shamrock 303i) and CCD (Andor Newton 920) for spectral analysis. For the PL measurements, a picosecond pulse laser with wavelength of 447 nm was used as excitation light and an objective (×40, NA = 0.6, Nikon) was employed to focus the excitation light on the sample and collect the epic scattered PL light. The same spectrometer and CCD combo as described above were used for PL spectral analysis.

**Transient absorption microscopy (TAM)**. The home-built TAM setup used in this study has been described in our previous publication[31]. Briefly, the output from two independent optical parametric amplifiers (OPAs, 400 kHz, ~300 fs, Light Conversion) served as pump and probe beam respectively, which were focused on sample with an objective (×60, NA = 0.95, Nikon). An acousto-optic modulator (Gooch and Housego, R23080-1) triggered by a lock-in amplifier (SR830, SRS, Inc.) was used to modulate the pump beam. A mechanical translation stage (Thorlabs, DDS600-E) was used to delay the probe with respect to the pump. A 2D Galvo mirror (Thorlabs GVS012) was employed to scan the probe beam relative to the pump beam in space (path 1 in Supplementary Fig. 6). Spatial filters were used to optimize the profile of the beams. The transmitted probe light was detected by an avalanche photodiode (APD) (Hamamatsu, C5331-04). The pump-induced change in the probe reflection ($\Delta T$) was extracted by the lock-in amplifier.

## Data availability

All data needed to evaluate the conclusions in the paper are present in the paper and/or the supplementary information. Correspondence and requests for materials should be addressed to L.H.

## Code availability

Correspondence and requests for codes used in the paper should be addressed to L.H.

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

## Acknowledgements

L.H. S.D., L.Y. and L.J. acknowledge the support from U.S. Department of Energy, Office of Basic Energy Sciences through award DE-SC0016356. L.D. acknowledges the support from U.S. Department of Defense, Office of Naval Research (Grant Number: N00014-19-1-2296, Program Manager: Dr. Paul Armistead and Dr. Joe Parker). E.S. acknowledges the support from Davidson School of Chemical Engineering of Purdue University.

## Author contributions

S.D. and L.H. designed the experiments. S.D., L.Y. and L.J. carried out the optical measurements. E.S. and L.D. contributed to sample growth and characterizations. S.D., L.Y. and L.H. analyzed experimental data. S.D. and L.H. wrote the manuscript with input from all authors.

## Competing interests

The authors declare no competing interests.
