## [Peer Review File · Nature Communications]

Reviewers' comments:

Reviewer #1 (Remarks to the Author):

In this manuscript entitled "Long-Range Exciton Transport and Slow Annihilation in Two-Dimensional Hybrid Perovskites", Shibin Deng, et al. investigated the exciton transport and annihilation in Ruddlesden-Popper phase hybrid perovskite by transient absorption microscopy (TAM). Exciton diffusion constant and annihilation rate have been derived by modeling the TAM results. The idea and the data collection and analysis are excellent. The manuscript is well organized and presented.

Some suggestions:

- 1) Crystal structure in Figure 1a should be checked where MA cation could not be found. And the position of long-chain BA cation should be checked.
- 2) In page-3, in the sentence 'The highly tunable structure coupled with near-unity ...', near-unity emission quantum yield is not suitable. In the reference [3], 79% PLQY is achieved, less than 90%.
- 3) In Figure 4e, the experiment data at ~ 100 ps is heavily deviated from the fitting line.
- 4) Spelling and grammar errors should be checked and corrected. For example, Line-7, page-2, 'reveling' should be 'unraveling' or 'revealing'; line-7, page-6, 'this is resulted from' should be corrected; line-11, page-7, the unit of exciton density ' $2.8 \times 10^{12} \text{ cm}^{-2}$ ' is not right.
- 5) Similar results in MAPbI₃ perovskite film measured with TAM technique have been reported in their previous paper (see Guo, Zhi, et al. "Spatial and temporal imaging of long-range charge transport in perovskite thin films by ultrafast microscopy." Nature communications 6 (2015): 7471). Although the two stories differ, the mechanism and results are quite similar. The authors should emphasize the new contributions. The authors also should compare their results with the previously reported charge carrier dynamics in two dimensional or multidimensional perovskites (such as Nature Communications 8, 14558 (2017)).

Reviewer #3 (Remarks to the Author):

Deng et al present an experimental investigation of exciton dynamics in 2D hybrid perovskites using transient absorption microscopy. The analysis of the data and the interpretation provide reinforcement to what is known about the resiliency of charge carriers to defects in hybrid perovskite materials. Additionally exciton-exciton annihilation rates are found to be small when compared to other 2D materials because of peculiar dielectric function of these polar materials that lead to polaronic effects. The paper is well written and the conclusion are backed up by analysis of the experimental data and I recommend for publication as is.

Reviewer #4 (Remarks to the Author):

The authors have investigated the diffusion and annihilation of excitons in 2D hybrid perovskites, which are very promising materials for solar cells and optoelectronics applications. The study is based on transient absorption microscopy. The authors report for the first time values for the diffusion constant and annihilation rates dependent on the thickness of the 2D perovskites, from a monolayer to 5 perovskite layers. The slow annihilation rates and relatively important diffusion lengths are very interesting for light emission devices. The samples are in the form of thin exfoliated crystals. The purity and quality of the samples are well characterized. The presentation of the data is clear and the manuscript well-written. The results will likely interest the broad audience of Nature Communications. I have the following comments:

- 1- The model of the authors take into account the diffusion of excitons, the radiative recombination

and the annihilation of excitons, and a third order Auger recombination due to the presence of free carriers as the number of layers increases. The model seems to give a clear evolution for all rates as function of the thickness except for k_3 , the third order rate, which appears to be maximum for $n=3$ and then counterintuitively decreases for $n=4, 5$. With higher n value, we may expect a larger fraction of free carriers. How the authors could explain this result? A fast defect-assisted recombination could also play a significant role in the dynamics?

2- The estimation of the transport properties of hybrid perovskites is very important but leads to a lot of discrepancies in the literature, regarding for example the diffusion coefficient with variations of several order of magnitudes. The authors cite a value of $1 \text{ cm}^2\text{s}^{-1}$ for the 3D limit. However, different studies point toward an overestimation of the diffusion coefficient due to photon transport and photon recycling. For example, the study by Bercegol et al. (Nature Communications, (2019) 10:1586 10.1038/s41467-019-09527) gives an estimation of only $D \approx 0.03 \text{ cm}^2 \text{ s}^{-1}$. I think that it is worth noting that photon recycling might impact the exciton density in 2D hybrid perovskites (see also Gan et al. Adv. Energy Mater. 10.1002/aenm.201900185) and the estimation of the diffusion constant in transient absorption microscopy.

Point-by-point response to the reviewers' comments

Reviewer #1

In this manuscript entitled “Long-Range Exciton Transport and Slow Annihilation in Two-Dimensional Hybrid Perovskites”, Shibin Deng, et al. investigated the exciton transport and annihilation in Ruddlesden-Popper phase hybrid perovskite by transient absorption microscopy (TAM). Exciton diffusion constant and annihilation rate have been derived by modeling the TAM results. The idea and the data collection and analysis are excellent. The manuscript is well organized and presented.

Response:

We appreciate greatly the reviewer's insights/suggestions for improving the manuscript.

Some suggestions:

Comment # 1: Crystal structure in Figure 1a should be checked where MA cation could not be found. And the position of long-chain BA cation should be checked.

Response:

We thank the reviewer for the suggestion. We have updated Fig. 1a based on the corresponding crystallographic information of $BA_2MA_{n-1}Pb_nI_{3n+1}$ ($n=1\sim 5$).

Comment # 2: In page-3, in the sentence ‘The highly tunable structure coupled with near-unity ...’, near-unity emission quantum yield is not suitable. In the reference [3], 79%

PLQY is achieved, less than 90%.

Response:

We agree with the reviewer that more accurate PL quantum yield should be given here. In the revised manuscript, we revised the sentence on page 3 as follows:

“The highly tunable structure coupled with emission quantum yield as high as 79% represents a significant advantage of 2D perovskites over other 2D semiconductors.”

Comment # 3: In Figure 4e, the experiment data at ~100 ps is heavily deviated from the fitting line.

Response:

We thank the reviewer for pointing this out.

The deviation of from the fitting for $n = 1$ can be explained by defect-assisted recombination that is not included in the model. An additional fast decay due to exciton trapping by defects can be observed when the exciton density is below the defect density. The defect density generally increases as n decreases. For example, the pump fluence necessary for filling the defects decreases from 10s of $\mu\text{J}\cdot\text{cm}^{-2}$ to below $1 \mu\text{J}\cdot\text{cm}^{-2}$ when n increases from 1 to 4. (Delpont, G. et al., Exciton-Exciton Annihilation in Two-dimensional Halide Perovskites at Room Temperature. Ref. 30). In our TAM measurements, fluences are ranging from 10s of $\mu\text{J}\cdot\text{cm}^{-2}$ to 100s of $\mu\text{J}\cdot\text{cm}^{-2}$, sufficiently high for filling the defects in $n > 1$. However, for $n = 1$, due to the higher defect density, not all the defects are passivated by the pump photons. These unpassivated defects lead to an additional fast decay in the first 100 ps for $n = 1$, which causes a deviation from the modeling by ~5%.

A short discussion has been added on pages 9-10 of the revised manuscript to address this comment,

“The model has satisfactorily fitted the experimental results with the initial exciton density varying over one order of magnitude using the same set of parameters for each n . The initial exciton densities were determined experimentally using absorption coefficient shown in Fig.1c. A deviation from the modeling by ~5% can be seen in the first 100 ps for $n = 1$, which is likely due to defect trapping of the excitons that is not included in the modeling. Defect density was the highest in $n = 1$, and probably not all the defects were passivated by the pump photons under our experimental conditions³⁰. For $n = 2-5$, such deviation was not observed because defect density decreased in thicker quantum wells and the pump fluences used were sufficiently high to passivate the defects³⁰.”

Comment # 4: Spelling and grammar errors should be checked and corrected. For example, Line-7, page-2, ‘reveling’ should be ‘unraveling’ or ‘revealing’; line-7, page-6, ‘this is resulted from’ should be corrected; line-11, page-7, the unit of exciton density ‘ 2.8×10^{12}

cm² is not right.

Response:

We thank the reviewer for pointing out these mistakes, we have corrected them as suggested. We have also proofread the revised manuscript.

Comment # 5: Similar results in MAPbI₃ perovskite film measured with TAM technique have been reported in their previous paper (see Guo, Zhi, et al. "Spatial and temporal imaging of long-range charge transport in perovskite thin films by ultrafast microscopy." Nature communications 6 (2015): 7471). Although the two stories differ, the mechanism and results are quite similar. The authors should emphasize the new contributions. The authors also should compare their results with the previously reported charge carrier dynamics in two dimensional or multidimensional perovskites (such as Nature Communications 8, 14558 (2017)).

Response:

We thank the reviewer for the suggestion. In the revised manuscript, we have highlighted the key contributions that distinct this manuscript from our previous work and the existing literature in the field.

The most important difference between our current work and our previous work is the central role of excitonic effects in 2D perovskites. In our previous work, the focus is on the transport of free charge carriers in bulk 3D perovskites. In contrast, due to the reduced dielectric screening, strongly bound excitons dominate the optical response in 2D perovskites. As a result, electron-hole interactions and dielectric screening play a key role in determining exciton transport and annihilation as demonstrated by the quantum-well-thickness-dependent studies presented here.

We have added these two sentences on page 4.

“We have demonstrated in a previous work the capability of TAM in directly imaging free carrier diffusion and extracting Auger recombination constants in bulk 3D perovskites³¹. Distinct from the results in 3D perovskites, the measurements on 2D perovskites elucidated the critical role of electron-hole interactions in controlling exciton dynamics and transport.”

We thank the reviewer for pointing out the reference (Xing et al., Nature Communications 8, 14558 (2017)). We have included this reference (Ref. 51) in the revised manuscript. We have also added one sentence on page 16: “This is consistent with a recent study⁵¹ in which van-der-Waals-coupled 2D perovskite multi-quantum-wells were shown to have better light emitting performance than 3D perovskites due to strongly bound excitons.”

Reviewer #3

Deng et al present an experimental investigation of exciton dynamics in 2D hybrid perovskites using transient absorption microscopy. The analysis of the data and the interpretation provide reinforcement to what is known about the resiliency of charge carriers to defects in hybrid perovskite materials. Additionally exciton-exciton annihilation rates are found to be small when compared to other 2D materials because of peculiar dielectric function of these polar materials that lead to polaronic effects. The paper is well written and the conclusions are backed up by analysis of the experimental data and I recommend for publication as is.

Response:

We thank the reviewer for the efforts in reviewing the manuscript and the comment about the quality of our work.

Reviewer #4

The authors have investigated the diffusion and annihilation of excitons in 2D hybrid perovskites, which are very promising materials for solar cells and optoelectronics applications. The study is based on transient absorption microscopy. The authors report for the first time values for the diffusion constant and annihilation rates dependent on the thickness of the 2D perovskites, from a monolayer to 5 perovskite layers. The slow annihilation rates and relatively important diffusion lengths are very interesting for light emission devices. The samples are in the form of thin exfoliated crystals. The purity and quality of the samples are well characterized. The presentation of the data is clear and the manuscript well-written. The results will likely interest the broad audience of Nature Communications.

Response:

We thank the reviewer for the comments and for his/her constructive suggestions for improving the manuscript.

I have the following comments:

Comment # 1: The model of the authors takes into account the diffusion of excitons, the radiative recombination and the annihilation of excitons, and a third order Auger recombination due to the presence of free carriers as the number of layers increases. The model seems to give a clear evolution for all rates as a function of the thickness except for k_3 , the third order rate, which appears to be maximum for $n=3$ and then counterintuitively decreases for $n=4, 5$. With higher n value, we may expect a larger fraction of free carriers.

How the authors could explain this result?

Response:

We thank the reviewer for pointing this out.

In the revised manuscript, we have added a short discussion on page 13 to address this comment.

“For Auger recombination of free carriers, a maximum is observed for k_3 when $n = 3$. The nonmonotonic n dependence of k_3 can be explained by the combination of the following two competing effects: (1) similar to EEA, the Auger recombination probability for free carriers per encounter decreases with increasing n due to enhanced screening; and (2) the fraction of free carriers increases with as n increases.”

*More specifically, Auger recombination probability per encounter decreases with increasing n because Coulomb interaction between free carriers is reduced by more effective dielectric screening in the thicker quantum-wells. This is similar to the reduced annihilation probability for the excitons in the thicker quantum-wells. Another reason for the increased Auger recombination rate at smaller n is the relaxed momentum conservation, which has been observed previously in quantum wells. (Dyakonov, M. I.; Kachorovskii, V. Y., Nonthreshold Auger recombination in quantum wells. *Physical Review B* **1994**, 49 (24), 17130-17138.)*

Comment # 3. A fast defect-assisted recombination could also play a significant role in the dynamics?

Response:

We agree with the reviewer that defect-assisted recombination pathways could play a role in the dynamics of 2D perovskites. Please also see our response to comment 3 of Reviewer #1.

A short discussion has been added on pages 9-10 of the revised manuscript to address this comment,

“The model has satisfactorily fitted the experimental results with the initial exciton density varying over one order of magnitude using the same set of parameters for each n . The initial exciton densities were determined experimentally using absorption coefficient shown in Fig.1c. A deviation from the modeling by ~5% can be seen in the first 100 ps for $n = 1$, which is likely due to defect trapping of the excitons that is not included in the modeling. Defect density is the highest in $n=1$, and probably not all the defects were passivated by the pump photons under our experimental conditions³⁰. For $n = 2-5$, such deviation was not observed because defect density decreases in thicker quantum wells and the pump fluences used were sufficiently high to passivate the defects³⁰.”

Comment # 4: The estimation of the transport properties of hybrid perovskites is very

important but leads to a lot of discrepancies in the literature, regarding for example the diffusion coefficient with variations of several order of magnitudes. The authors cite a value of $1 \text{ cm}^2\text{s}^{-1}$ for the 3D limit. However, different studies point toward an overestimation of the diffusion coefficient due to photon transport and photon recycling. For example, the study by Bercegol et al. (Nature Communications, (2019) 10:1586 10.1038/s41467-019-09527) gives an estimation of only $D \approx 0.03 \text{ cm}^2 \text{ s}^{-1}$. I think that it is worth noting that photon recycling might impact the exciton density in 2D hybrid perovskites (see also Gan et al. Adv. Energy Mater. 10.1002/aenm.201900185) and the estimation of the diffusion constant in transient absorption microscopy.

Response:

We thank the reviewer for bringing this point up and providing the references (they are included as Ref.41 and 42). We agree with the reviewer that photon recycling could lead to an overestimation of the diffusion constants measured by TAM.

To address this comment, we have carried out Monte Carlo simulations to differentiate electronic transport from photonic transport. The results are shown in the new Supplementary Fig. 9 (also shown below) and detailed in Supplementary Note 4.

Supplementary Fig. 9. Monte Carlo simulation of carrier transport. a-c) simulated TAM profile for electronic transport (a), photonic transport (b) and experimental TAM profile (c), the curves in (a) and (b) are normalized to its maximum, the curves in (c) are normalized to the maximum of the curve at 2 ps. d-f) the residual of Gaussian fitting for (a-c).

The following paragraph has been added on pages 11-12 of the revised manuscript.

“It is possible that photon recycling that has been proposed to enhance carrier diffusion in 3D perovskites contributes to the long-range exciton transport observed here⁴⁰⁻⁴². To

differentiate the contributions from photon recycling from pure exciton diffusion, we have carried out Monte Carlo simulations as shown in Supplementary Fig. 9. As detailed in Supplementary Note 3, both electronic transport and photonic transport from photon recycling can be microscopically described as random walks but with significantly different parameters (Supplementary Table 1). For electronic transport, each step of the random walk during τ_s between scattering events is on the order of nm. To achieve a final average transport distance of 100s of nm, excitons need to walk $\sim 10^5$ steps randomly, resulting in a Gaussian distribution of transport distance. For photonic transport, the step size is much larger, on the order of 100s of nm as determined by the penetration depth $1/\alpha$. To achieve a final average transport distance of 100s of nm, the number of steps for photons is small, such as 1 or 2, which results in a distribution of transport distance strongly deviates from the Gaussian type. Therefore, we can use the TAM profiles to determine if transport via photon recycling playing a role by examining the spatial distributions of the excitons. As shown in Supplementary Fig. 9, the exciton population profiles are described very well by Gaussian functions. Therefore, we conclude that photonic contributions from photon recycling in exciton transport is negligible.”

As shown in Supplementary Fig. 9, we replot the TAM profiles in Figure 2b and in Supplementary Fig. 9c. The fitting residuals of the TAM results (Supplementary Fig. 8f) are unstructured which means the spatial distributions of excitons is Gaussian type, suggesting negligible contribution from photonic transport.

Finally, the photonic contributions can be highly affected by (1) PLQY; (2) geometry of the sample; and (3) dielectric mismatch between sample and environment. Under the right conditions, we expect exciton transport can be enhanced using photonic mechanism over the intrinsic electronic transport.

Additional changes:

We have included sensitivity analysis of the model to the fitting parameters in Supplementary Fig. 8.

REVIEWERS' COMMENTS:

Reviewer #1 (Remarks to the Author):

The revised version of the manuscript titled "long-range exciton transport and slow annihilation in two-dimensional hybrid perovskites" has addressed the previously raised questions. The overall quality of the manuscript in terms of content and scientific presentation has improved to a point that allows for publication in Nature Communications.

Reviewer #4 (Remarks to the Author):

The authors have answered all my remarks and improved their manuscript. I recommend it for publication in Nature Communications

Point-by-point response to the reviewers' comments

Reviewer #1

The revised version of the manuscript titled "long-range exciton transport and slow annihilation in two-dimensional hybrid perovskites" has addressed the previously raised questions. The overall quality of the manuscript in terms of content and scientific presentation has improved to a point that allows for publication in Nature Communications.

Response:

We thank the reviewer for the efforts in reviewing the manuscript and the comment about the quality of our work.

Reviewer #4

The authors have answered all my remarks and improved their manuscript. I recommend it for publication in Nature Communications

Response:

We thank the reviewer for the efforts in reviewing the manuscript and the comment about the quality of our work.